# Intracellular Aluminium in Inflammatory and Glial Cells in Cerebral Amyloid Angiopathy: A Case Report

**DOI:** 10.3390/ijerph16081459

**Published:** 2019-04-24

**Authors:** Matthew Mold, Jason Cottle, Andrew King, Christopher Exley

**Affiliations:** 1The Birchall Centre, Lennard-Jones Laboratories, Keele University, Staffordshire ST5 5BG, UK; m.j.mold@keele.ac.uk; 2School of Medicine, David Weatherly Building, Keele University, Staffordshire ST5 5BG, UK; jasoncottlee@gmail.com; 3Department of Clinical Neuropathology, Kings College Hospital, London SE5 9RS, UK; andrewking@nhs.net

**Keywords:** cerebral amyloid angiopathy, brain aluminium, pro-inflammatory cells, human exposure to aluminium, Camelford in Cornwall

## Abstract

(1) Introduction: In 2006, we reported on very high levels of aluminium in brain tissue in an unusual case of cerebral amyloid angiopathy (CAA). The individual concerned had been exposed to extremely high levels of aluminium in their potable water due to a notorious pollution incident in Camelford, Cornwall, in the United Kingdom. The recent development of aluminium-specific fluorescence microscopy has now allowed for the location of aluminium in this brain to be identified. (2) Case Summary: We used aluminium-specific fluorescence microscopy in parallel with Congo red staining and polarised light to identify the location of aluminium and amyloid in brain tissue from an individual who had died from a rare and unusual case of CAA. Aluminium was almost exclusively intracellular and predominantly in inflammatory and glial cells including microglia, astrocytes, lymphocytes and cells lining the choroid plexus. Complementary staining with Congo red demonstrated that aluminium and amyloid were not co-located in these tissues. (3) Discussion: The observation of predominantly intracellular aluminium in these tissues was novel and something similar has only previously been observed in cases of autism. The results suggest a strong inflammatory component in this case and support a role for aluminium in this rare and unusual case of CAA.

## 1. Introduction

Human exposure to aluminium is burgeoning with consequent implications for health [1]. The neurotoxicity of aluminium in humans is well established, for example in dialysis encephalopathy [2], while the role of aluminium in other neurodegenerative diseases remains to be confirmed [3]. The aluminium content of human brain tissue has been reviewed recently [4] and subsequent studies have provided data on aluminium in human brain tissue in familial Alzheimer’s disease [5], autism [6] and multiple sclerosis [7]. Unequivocal quantitative analyses of aluminium in human brain tissue are now providing a strong basis for a role of aluminium in brain disease [8,9]. The recent development of aluminium-specific fluorescence microscopy now provides a relatively simple and unequivocal method of visualising aluminium in human brain tissue [10]. It has been used successfully to identify the location of aluminium in brain tissue in sporadic Alzheimer’s disease, familial Alzheimer’s disease, autism and multiple sclerosis [5,6,7,10]. Herein, we returned to a case of cerebral amyloid angiopathy (CAA) with coincident very high levels of brain aluminium [11] to identify the location of aluminium in these tissues using aluminium-specific fluorescence microscopy. The results are perhaps surprising in light of what was originally proposed.

## 2. Case Summary

In 2006 we reported on a case of severe CAA with coincident high levels of brain aluminium [11]. The case was of a woman who in 1998 had been exposed to very high and sustained levels of aluminium in her water supply, a notorious water pollution incident in the Cornish town of Camelford. The woman died of what was described as a rare form of sporadic early onset β amyloid angiopathy in cerebral cortical and leptomeningeal vessels. She also had limbic stage Lewy body pathology. Full details of the neuropathology in this case have been described elsewhere [11]. The role played by coincident very high levels of aluminium in affected regions of the cortex was unknown, though an association with amyloid deposits was suggested. Here, we used aluminium-specific fluorescence microscopy to identify the location of aluminium in these brain tissues to test the suggestion of an association with the deposition of β amyloid.

All chemicals were from Sigma Aldrich, Gillingham, UK unless otherwise stated. Ethical approval along with tissues was obtained from the Oxford Brain Bank (15/SC/0639). Numbered adjacent serial sections were prepared at a thickness of 5 μm on glass slides from paraffin-embedded brain tissue blocks of the frontal, parietal, temporal and occipital lobes, and of the hippocampus. Lumogallion (4-chloro-3-(2,4-dihydroxyphenylazo)-2-hydroxybenzene-1-sulphonic acid, TCI Europe N.V., Belgium) staining was performed as previously described [6,10]. Briefly, sections were de-waxed via transfer into fresh Histo-Clear (National Diagnostics, US) and rehydrated through an ethanol (HPLC grade) gradient from 100 to 30% v/v prior to rehydration in ultrapure water (cond. <0.067 μS/cm). Staining was performed in moisture chambers in which rehydrated sections were outlined with a hydrophobic PAP pen to which 200 μL of 1 mM lumogallion in 50 mM PIPES, pH 7.40, was added, or buffer only was used for non-stained autofluorescence sections. Sections were incubated at ambient temperature for 45 min away from light, prior to rinsing in PIPES buffer and mounting with Fluoromount™. Congo red staining was performed via the immersion of rehydrated sections into 0.5% w/v Congo red in 50% v/v ethanol for 5 min. Sections were differentiated in a solution of 0.2% w/v potassium hydroxide in 80% v/v ethanol for 3 s and rinsed in ultrapure water for 30 s. Sections prepared in this way were mounted with Faramount (Dako, UK).

All microscopy analyses were performed using an Olympus BX50 fluorescence microscope, equipped with a vertical illuminator and BX-FLA reflected light fluorescence attachment (mercury source). Lumogallion fluorescence, characteristically bright orange to intense yellow depending upon the concentration of available Al^3+^, and respective autofluorescence in the absence of the fluorophore were visualised using a U-MNIB3 (excitation filter: 470–495 nm; dichromatic mirror: 505 nm; longpass emission filter: 510 nm) fluorescence filter cube (Olympus, UK). Light exposure and transmission settings were fixed across respective staining conditions. Brain tissue sections (Figure 1, Figure 2, Figure 3 and Figure 4) were sequentially scanned for the identification of lumogallion-reactive aluminium and where positive fluorescence was identified and complementary autofluorescence was assessed on a non-stained serial section. Apple-green birefringence of Congo red stained amyloid deposits was obtained using a U-POT drop-in polariser and a U-ANT transmitted light analyser (both from Olympus, UK) and assessed through sequential screening. All images were acquired using the CellD software suite (Olympus, SiS Imaging Solutions, SiS, GmbH). Bright-field and fluorescence channels were merged using Photoshop (Adobe Systems, Inc., US).

In the parietal lobe, aluminium was found predominantly in white matter (Figure 1, regions 1, 2 and 4) and almost exclusively as intracellular deposits in glial-like cells associated with vasculature and cell debris (Figure 5a,b). Intraneuronal aluminium was identified in an area at a grey–white matter interface (Figure 1, region 3). Serial non-stained sections for autofluorescence confirmed the presence of aluminium in each of the identified regions (Appendix A). Severe CAA was widespread in leptomeningeal vessels (Figure 1, region 6) as revealed by apple-green birefringence under fully polarised light. Maltese cross diffraction patterns revealed spherulites in this tissue (Figure 1, region 5).

In the occipital lobe, only intracellular aluminium was identified in both white (Figure 2, regions 1, 2 and 6) and grey matter (Figure 2, regions 3, 4 and 5). Aluminium-loaded microglial-like cells were observed surrounding astrocytes in grey matter (Figure 6a) while aluminium in astrocytes within white matter appeared as punctate deposits, approximately 1 μm in diameter (Figure 6b). Serial non-stained sections for autofluorescence confirmed the presence of aluminium in each of the identified regions (Appendix A). Spherulites, with occasional apple-green birefringence, were identified in grey matter (Figure 2, regions 7 and 8).

Aluminium was abundant in grey and white matter of the temporal lobe and predominantly intracellular (Figure 3, regions 1, 2, 3, 4, 6 and 8) as opposed to extracellular (Figure 3, regions 5 and 7). Intracellular deposits were found in glial cells exhibiting astrocytic-like processes (Figure 7). Serial non-stained sections for autofluorescence confirmed the presence of aluminium in each of the identified regions (Appendix A). Extensive CAA was evident (Figure 3, regions 9 and 10) and exhibited the classical apple-green birefringence characteristic of amyloid (Figure 9).

Intracellular aluminium in hippocampal tissue was predominantly found in white matter or associated with the grey-white matter interface (Figure 4, regions 1, 3, 4, 6 and 8). Extracellular deposits of aluminium appeared only in white matter (Figure 4, regions 2 and 5). Intracellular aluminium was found in inflammatory cells within cortical vessels (Figure 8a) and in cells lining the choroid plexus (Figure 8b). Serial non-stained sections for autofluorescence confirmed the presence of aluminium in each of the identified regions (Appendix A). Congo red positive staining was not observed in the hippocampus (Figure 4).

## 3. Discussion

Application of aluminium-specific fluorescence microscopy [10] confirmed the high content of aluminium in brain tissue in this case [11], as previously reported. A surprising feature was that deposits of aluminium were predominantly intracellular and located in grey and white matter across all four main lobes and in the hippocampus (Figure 1, Figure 2, Figure 3 and Figure 4). This is unusual because previous research identifying the location of aluminium in brain tissue in Alzheimer’s disease showed aluminium to be predominantly extracellular and associated with cellular debris [5]. This predominantly extracellular location of brain tissue aluminium was also observed in the only other ‘Camelford brain’ investigated to date [10]—a case of sporadic, early-onset Alzheimer’s disease that was also characterised by some unusual neuropathology [12]. 

While some aluminium was intraneuronal (Figure 1), most aluminium deposits were identified in non-neuronal cells including what appeared to be microglia, astrocytes, lymphocyte-like cells and cells lining the choroid plexus (Figure 5, Figure 6, Figure 7, Figure 8 and Figure 9). The predominantly intracellular and non-neuronal distribution of aluminium observed herein has previously only been observed in brain tissue in individuals who died with a diagnosis of autism [6].

We also confirmed, using Congo red and polarised light, the previous observation of severe, sporadic β amyloid angiopathy. Amyloid was not deposited as senile plaques, though numerous spherulites showing apple-green birefringence were identified [13]. We had expected to find the co-localisation of aluminium and β amyloid angiopathy in these tissues but, without exception, this was not the case. Aluminium and amyloid deposits were distinct from one another with no clear examples of co-localisation (Figure 9).

These are the first images of aluminium in brain tissue in a known case of CAA without significant Alzheimer’s disease pathology. The predominance of aluminium in microglia, astrocytes, lymphocytes and cells lining the choroid plexus has all the hallmarks of an inflammatory condition and supports recent observations of CAA associated with inflammation [14]. It may also support a peripheral, outside of the central nervous system, origin for CAA [15] in this case. While we know that the individual in this case was exposed to high levels of aluminium in their environment over an extended time period, we do not know if they suffered from a high body burden of aluminium. The very high content of aluminium in their brain tissue does support a high body burden and its presence in inflammatory cells in brain tissue and the vasculature offers the possibility that aluminium was carried into the brain concomitant with brain inflammation. The origin of putative brain inflammation in this case is unknown but could also be attributed to aluminium. The evidence presented herein supports the conclusion of the coroner in this case that aluminium was a likely contributor to her death.

There have been only two investigations of the neuropathology of brain tissue from individuals who were exposed to aluminium in their potable water at Camelford [10,11,12]. Both cases are characterised by unusual, if distinctly different, neuropathology and by decidedly different locations of coincident high levels of brain aluminium. While no conclusions can be drawn from just two cases, examinations of other Camelford cases, should brain tissue become available, are surely warranted.

## Figures and Tables

**Figure 1 ijerph-16-01459-f001:**
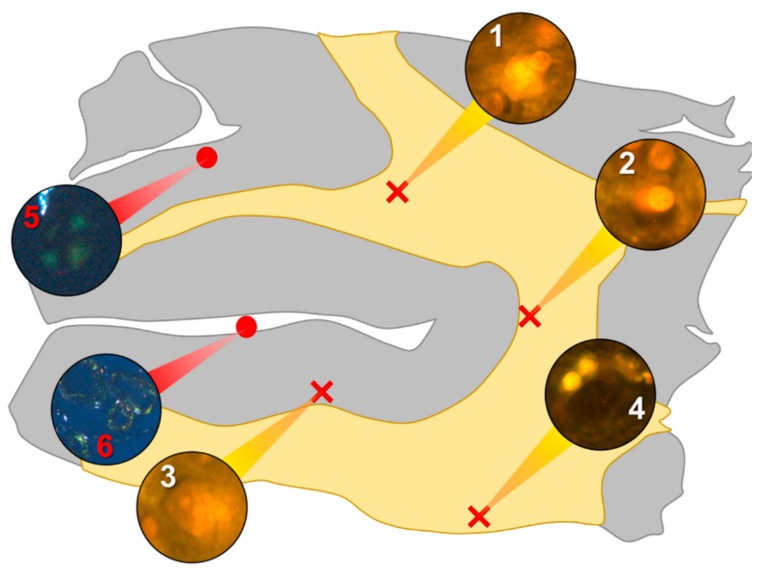
Schematic depicting a 5 μm tissue section of the parietal lobe. White matter and grey matter are highlighted yellow and grey, respectively. Lumogallion-reactive aluminium was identified through sequential scanning of a 5 μm tissue section, with positive bright orange fluorescence denoted by red crosses (regions 1–4). On an adjacent serial section, Congo red positive regions showing apple-green birefringence under polarised light identified amyloid and spherulites, denoted by red circles (regions 5 and 6).

**Figure 2 ijerph-16-01459-f002:**
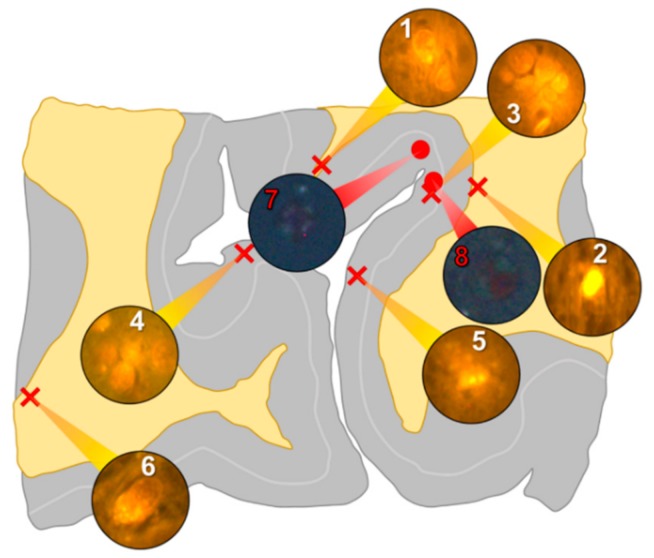
Schematic depicting a 5 μm tissue section of the occipital lobe. White matter and grey matter are highlighted yellow and grey, respectively. Lumogallion-reactive aluminium was identified through sequential scanning of a 5 μm tissue section, with positive bright orange fluorescence denoted by red crosses (regions 1–6). On an adjacent serial section, Congo red positive regions showing apple-green birefringence under polarised light identified amyloid and spherulites, denoted by red circles (regions 7 and 8).

**Figure 3 ijerph-16-01459-f003:**
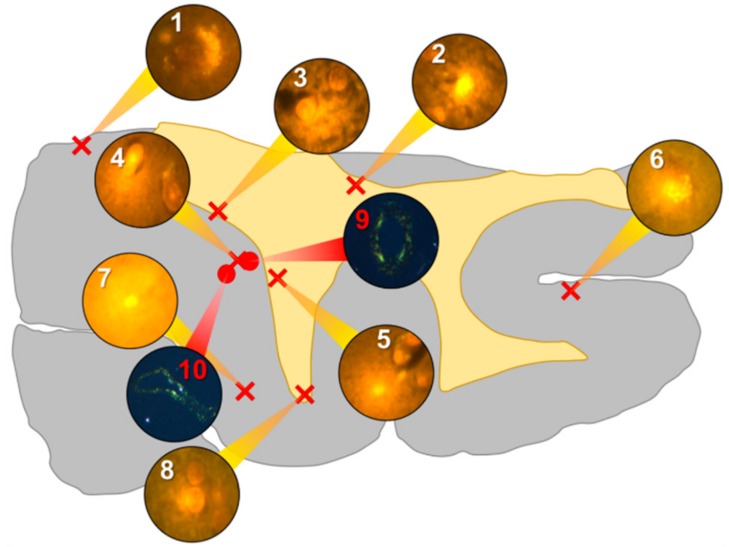
Schematic depicting a 5 μm tissue section of the temporal lobe. White matter and grey matter are highlighted yellow and grey, respectively. Lumogallion-reactive aluminium was identified through sequential scanning of a 5 μm tissue section with positive bright orange fluorescence denoted by red crosses (regions 1–8). On an adjacent serial section, Congo red positive regions showing apple-green birefringence under polarised light identified amyloid, denoted by red circles (regions 9 and 10).

**Figure 4 ijerph-16-01459-f004:**
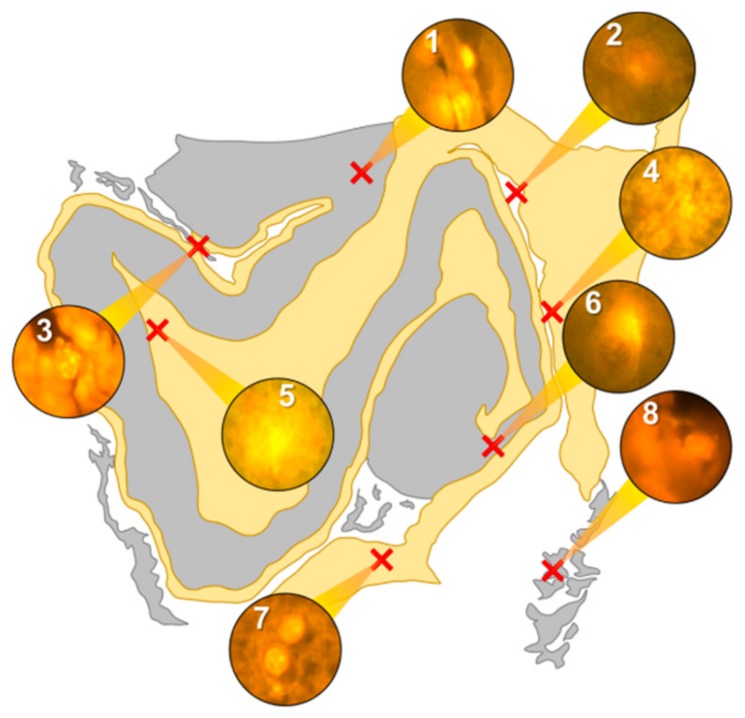
Schematic depicting a 5 μm tissue section of the hippocampus. White matter and grey matter are highlighted yellow and grey, respectively. Lumogallion-reactive aluminium was identified through sequential scanning of a 5 μm tissue section, with positive bright orange fluorescence denoted by red crosses (1–8).

**Figure 5 ijerph-16-01459-f005:**
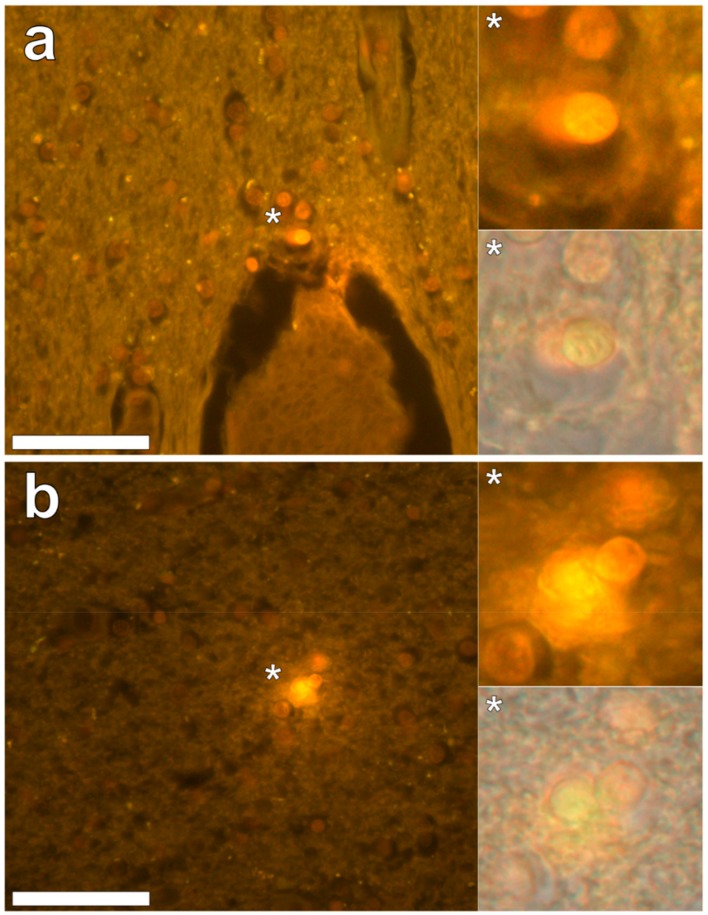
Lumogallion-reactive aluminium in glial cells in white matter of the parietal lobe. Intracellular bright orange fluorescence was noted in glial cells surrounding vasculature (**a**) and in areas depicting cellular debris (**b**). Magnified inserts are denoted by asterisks with lower panels including a bright field overlay. Magnification 400×; scale bars 50 μm.

**Figure 6 ijerph-16-01459-f006:**
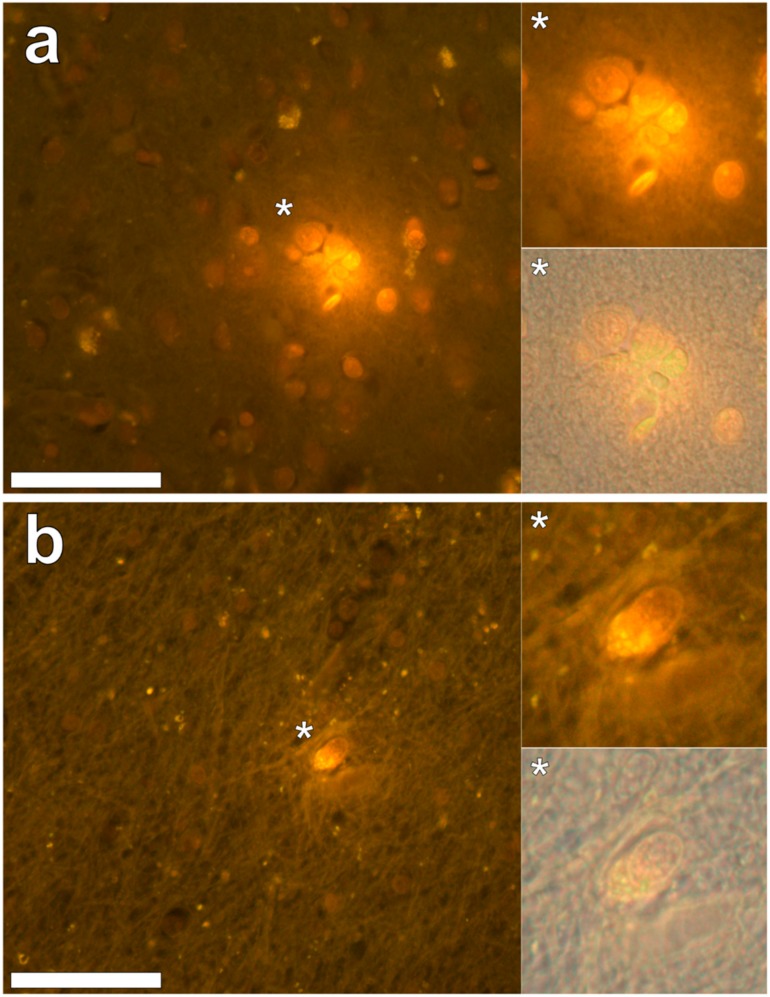
Intracellular lumogallion-reactive aluminium in glial cells in grey and white matter of the occipital lobe. Cells morphologically compatible with microglia surrounding astrocytes (**a**) and an aluminium loaded astrocytic-like cell, exhibiting intracellular bright orange fluorescence (**b**), were identified in grey and white matter regions, respectively. Magnified inserts are denoted by asterisks with lower panels including a bright field overlay. Magnification 400×; scale bars 50 μm.

**Figure 7 ijerph-16-01459-f007:**
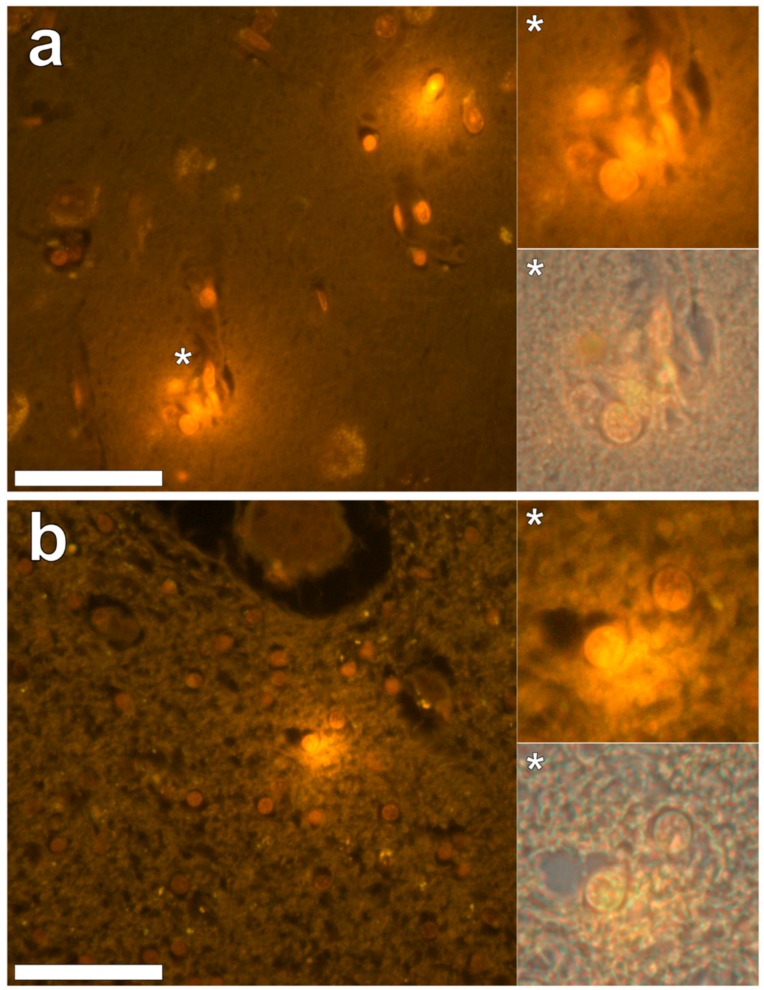
Intracellular lumogallion-reactive aluminium identified in grey and white matter regions of the temporal lobe. Glial cells exhibiting astrocytic-like processes displayed bright orange fluorescence in grey (**a**) and white (**b**) matter regions. Magnified inserts are denoted by asterisks with lower panels including a bright field overlay. Magnification 400×; scale bars 50 μm.

**Figure 8 ijerph-16-01459-f008:**
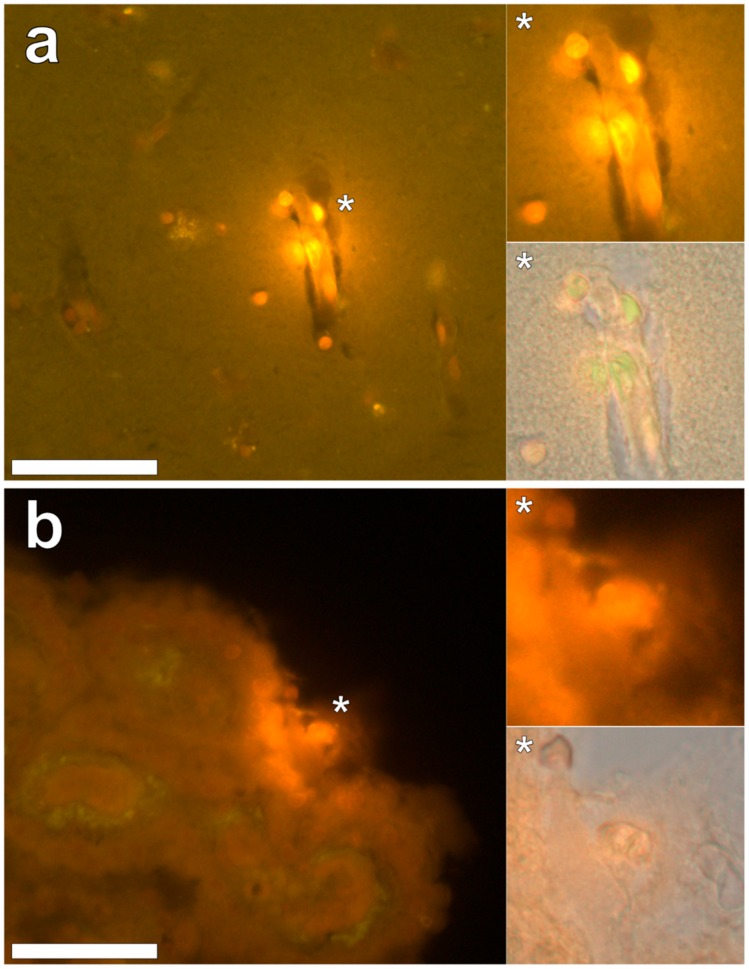
Intracellular lumogallion-reactive aluminium localised within the hippocampus. Intracellular bright orange fluorescence was noted in inflammatory cells in the vessel wall (**a**) and within ependymal cells lining the choroid plexus (**b**). Magnified inserts are denoted by asterisks with lower panels including a bright field overlay. Magnification 400×; scale bars 50 μm.

**Figure 9 ijerph-16-01459-f009:**
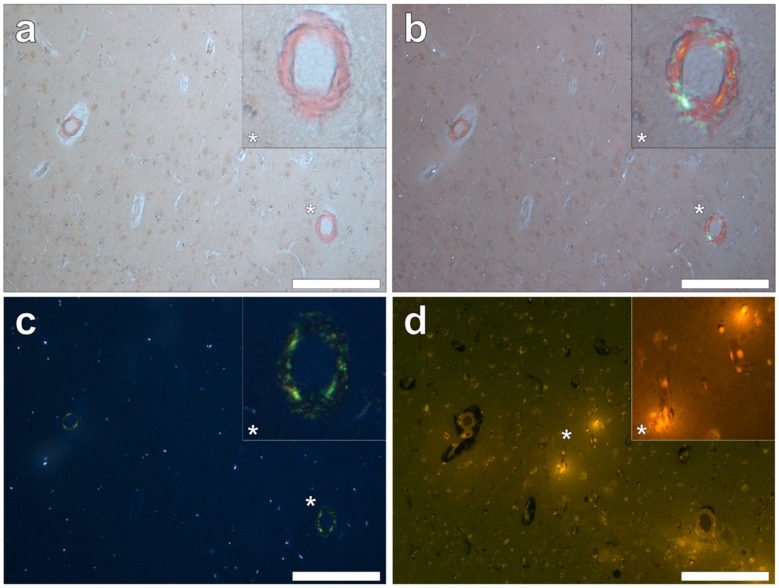
Congo red reactive amyloid deposited independently of lumogallion-reactive aluminium in the temporal lobe. Positive Congo red staining was observed under bright field (**a**), partial (**b**) and fully (**c**) polarised light, demonstrating an apple-green birefringence confirming the presence of amyloid with a β-pleated sheet conformation. Intracellular aluminium identified in glial-like cells (**d**) was not co-located with amyloid within the vasculature. Magnified inserts are denoted by asterisks. Magnification 100×; scale bars 200 μm.

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
