# Peer review of "Intracellular Aluminium in Inflammatory and Glial Cells in Cerebral Amyloid Angiopathy: A Case Report"

_ijerph, 2019, doi:10.3390/ijerph16081459_

Round 1

Reviewer 1 Report

Although the schematics are nicely designed,  it would be better to also show the actual tissue slide, for a complete overview of the deposits pattern.

I do miss some controls in this study design. For instance, would be ideal to compare CAA deposits vs AD plaques (authors published previously a work on AD, so it should not be too difficult for them to provide this comparison). Authors state that this particular type of deposition is peculiar in its localization, but I did miss to find colocalization of aluminum and amyloid in their previous work as well. 

Author Response

Please find our reply on the attached file.

It may be worth reiterating here that the unique and surprising obervation in this case study was the LACK of co-localisation of aluminium and amyloid.

Reviewer 2 Report

This is an interesting case report documenting the presence of aluminum in the brain of an individual exposed to high concentration of the metal in the drinking water. The main concern with the results is the lack of clear identification of the cell types where aluminum may be deposited.  Although the authors indicate that the aluminum appears to be in 'glial-like' or 'astrocyte-like' cells, clear identification of these cell types is warranted.  Thus it is necessary for the authors to conduct GFAP staining or cd11b staining to identify astrocytes and microglia respectively.  Furthermore, it should be determined if the aluminum located in the white matter is perhaps present in oligodendrocytes.  Only if the cells are definitively identified, can it be concluded that the aluminum is principally located in immune-competent cells of the CNS.   

Author Response

Please find our reply in the attached file.

I would like to point out that we do agree with the main criticism of the referee that unequivocal identification of different cell types within tissues is optimal. We have explained why this was not possible herein and we have added that one of us is an established neuropathologist, being THE neuropathologist for the London Brain Bank and so we rely upon his extremely good judgement.
